# The role of migration barriers for dispersion of Proliferative Kidney Disease—Balance between disease emergence and habitat connectivity

Heike Schmidt-Posthaus[1]*, Ernst Schneider[2], Nils Schölzel[3], Regula Hirschi[1], Moritz Stelzer[1], Armin Peter[3]

1 Centre for Fish and Wildlife Health, Department of Infectious Diseases and Pathobiology, Vetsuisse Faculty, University of Bern, Bern, Switzerland, 2 Thalwil, Switzerland, 3 FishConsulting GmbH, Olten, Switzerland

* heike.schmidt@vetsuisse.unibe.ch

**Data Availability Statement:** The data underlying this study are available on OSF (DOI: osf.io/ftb7m).

## Abstract

Natural and uninterrupted water courses are important for biodiversity and fish population stability. Nowadays, many streams and rivers are obstructed by artificial migration barriers, often preventing the migration of fish. On the other hand, distribution of pathogens by migrating fishes is still a point of concern. Pathogen transport and transmission is a driving force in the dynamics of many infectious diseases. The aim of the study was to investigate the possible consequences of the removal of an artificial migration barrier for the upstream transport of *Tetracapsuloides bryosalmonae*, the causative agent of Proliferative Kidney Disease (PKD) in brown trout, by migrating fish. To test this question, a river system was selected with a migration barrier separating a PKD positive river from a PKD negative tributary. After removal of the barrier, PKD prevalence and pathology was examined during five years after elimination of the barrier. In the tributary, no PKD was recorded at any time of the survey. By means of unidirectional PIT (passive integrated transponder)-tagging, we confirmed upstream migration of adult brown trout into the tributary during the cold season, presumably for spawning. By eDNA, we confirmed presence of *T. bryoalmonae* and *Fredericella sp.*, the definitive host, DNA in water from the PKD positive river stretch, but not in the PKD negative tributary. Our study illustrates the importance of the connectivity of streams for habitat maintenance. Although migration of brown trout from a PKD-positive river into a PKD-negative tributary, mainly for spawning, was confirmed, upstream spreading of PKD was not observed.

## Introduction

Water courses are important components of natural landscapes as well as a vital part of ecosystems. Many streams and rivers are obstructed and migration barriers prevent migration of fishes. For decades, free flowing rivers in Central Europe were seen as a threat rather than as a benefit, despite such areas often compromising the only functioning reservoirs of biodiversity. Fragmentation of river networks can increase the isolation of fish populations [1–4]. Species like brown trout

**Funding:** This study was supported by the Federal Office for the Environment (FOEN), Bern, Switzerland (HS) (www.bafu.admin.ch) in the form of materials contributed to HS and AP and part of a salary for HS. Additionally, the study was supported by FishConsulting GmbH in the form of salaries for NS and AP. The specific roles of these authors are articulated in the 'author contributions' section. FOEN had no role in the study design, data collection and analysis, decision to publish, or preparation of the manuscript. FishConsulting GmbH was involved in data collection and analysis and preparation of the manuscript, but had no role in the study design or decision to publish.

**Competing interests:** The authors have read the journal's policy and have the following competing interests: NS and AP are employees of FishConsulting GmbH, which was involved in data collection and analysis and preparation of the manuscript. This does not alter our adherence to PLOS ONE policies on sharing data and materials. There are no patents, products in development or marketed products associated with this research to declare.

(*Salmo trutta*) or salmon (*Salmo salar*) are particularly sensitive to barriers. Juvenile and adult life stages of these species must migrate extensively within freshwater river systems. Barriers are therefore a potentially important constraint on reproduction and population stability where access to and from spawning and rearing habitats is limited, delayed, or even barred [5]. Nowadays, the restoration of the water courses is included as a dynamic part of several projects, which propose full connectivity for migrating fish and expansion of water space [6].

On the other hand, distribution of pathogens by migrating fishes is still a point of concern. Pathogen transport by hosts is a driving force in the dynamics of many infectious diseases [7]. Proliferative Kidney Disease (PKD) in brown trout is one example where controversial arguments were put forward, weighting interests between habitat upgrade and dispersion of this devastating disease. PKD is an emerging disease in Europe [8] with high losses in farmed rainbow trout (*Oncorhynchus mykiss*) [9, 10] and wild salmonid populations [11–13]. Likely, the disease is one factor for the long term population decline of wild brown trout (*Salmo trutta*) in many Swiss midland river systems [14] and in neighboring Southern Germany [15]. PKD is caused by *Tetracapsuloides bryosalmonae*, belonging to the Malacospora, Cnidaria [16–20]. The life cycle of *T. bryosalmonae* involves bryozoans, mainly *Fredericella sultana*, as the definitive host [18, 21, 22] and salmonids as the intermediate host [17, 23]. Infected bryozoans release malacospores that infect salmonids by penetrating the gill epithelium and vascular walls. Through haematogenic dispersion, a generalized infection occurs, with the kidney as main target organ [24]. In the renal interstitial tissue, the parasite develops from presporogonic into sporogonic stages, which migrate into the tubular lumen and are eventually excreted via the urine [25, 26]. These fish malacospores infect susceptible bryozoans [27–29] to complete the parasite cycle. In the salmonid host, mainly young-of-the-year (YOY) show signs of disease, when the wild fish are exposed to the water-borne bryozoan malacospores for the first time in their life [8]. Following infection, brown trout can release spores up to five years post exposure [30].

PKD is a temperature dependent disease [26, 31], with a peak in disease severity at temperature above 15°C, corresponding with high temperatures typical of late summer / early autumn [32]. During winter and spring, the infection is usually inactive and asymptomatic [33]. However, Gay et al. (2001) [34] showed that infectious stages of the parasite can be present throughout the year.

The aim of this study was to investigate possible consequences of the removal of an artificial migration barrier for the upstream transport of *T. bryosalmonae* by migrating brown trout. To investigate this aim, a river system was selected with a migration barrier separating a PKD positive river, the Wutach, from a PKD negative tributary, the Ehrenbach. The removed barrier was situated close to the estuary of the Ehrenbach into the Wutach.

PKD prevalence and pathology in YOY brown trout were examined immediately before and over a period of five years after elimination of the barrier. To investigate upstream migration of brown trout from the PKD positive river stretches of the Wutach into the Ehrenbach, PIT (passive integrated transponder)-tagging of wild trout from the Wutach was used. In the Wutach, different age classes of brown trout (from YOY up to adult fish) were PIT-tagged. In addition, water samples were taken and filtered in the field to investigate the presence of *F. sultana* and *T. bryosalmonea* DNA to confirm the requirements for an entire parasite cycle.

## Material and methods

### Ethics statement

This study was carried out in accordance with the University of Bern Animal Ethics Committee. Approval for animal experiments was obtained from the Cantonal Veterinary Offices in Bern, Zürich and Schaffhausen, Switzerland (Authorization # BE102/14+ and # 75698).

## Study sites, fish sampling and eDNA sampling

The river Wutach and its tributary Ehrenbach are situated at the border between Southern Germany and the North-eastern part of Switzerland (Fig 1). The selection of the river system was based on previous investigations which showed a negative PKD status in brown trout in the Ehrenbach, but high infection prevalence in brown trout in the Wutach, ranging from 90 to 100% [35]. Beginning in 2013 (14[th] of September), 20 YOY brown trout were sampled above (Ehrenbach) and below the migration barrier (Wutach) to monitor the PKD status, as a reference sampling, before removal of the barrier. In the Wutach, two sampling sites were selected, one site above the estuary of the Ehrenbach and one below the estuary to monitor the PKD status of possibly migrating brown trout.

The removed migration barrier consisted of a 4.3 m wide and 1 m high stone wall, which was anchored in the stream bank with longitudinal constructions of the same material. The bottom of the natural riverbed ended at the upper edge of the wall. There was no flow through or around the barrier, a deep pool at the foot of the wall was present. This barrier prevented upstream movement of fish. Mid-June 2014, the barrier was dismantled and replaced by a 25 m long rough ramp with a gradient not higher than 1:15. The removal of the barrier was part of a restoration program (https://plattform-renaturierung.ch/). Starting from 2016 until 2019, the PKD status of resident brown trout YOY was evaluated. Samples were collected annually by electrofishing, same day from the Wutach and the Ehrenbach, between 24[th] of August and 15[th] of September 2016, 2017, 2018 and 2019. This sampling period was selected due to presenting the highest probability to detect infection with *T. bryosalmonae*.

Brown trout were captured by electrofishing. Two stretches of 100 m in the river Wutach (2'677'061.2/1'289'694.7, 2'677'814.2/1'291'059.7) and one of 50 m in the Ehrenbach (2'674'786/1'292'254), approximately 4 km upstream from the removed migration barrier were investigated (Fig 1). Twenty YOY brown trout from each of the three sampling sites were captured and euthanized separately by 150 mg/L buffered tricaine methanesulfonate (MS 222®, Argent Chemical Laboratories, Redmont, USA).

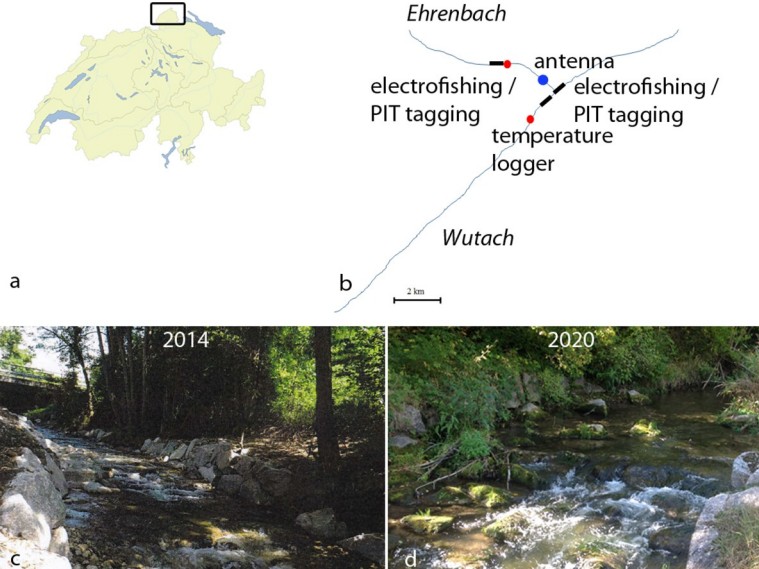

**Fig 1. Scheme of study area, photographs of area of former migration barrier.** a. Map of Switzerland, square marks location of study area; b. Scheme of study area shows locations of electrofishing, PIT-tagging, monitoring antenna and temperature loggers; c. Area of removed barrier in 2014, directly after removal; d. Same area in 2020.

In 2020, at each of the three locations in the Wutach and the Ehrenbach, respectively, three replicates of 300 ml of water each (nine water samples in total) were taken from the river banks and filtered immediately on site with a 50/60 ml BD Plastipak syringe (Becton Dickinson GmbH, Germany) and Sterivex[TM]- filters (Merck Millipore[®], Merck, Germany) of 0.45μm pore size. Filters were put on ice for transport and stored at -20˚C until DNA extraction.

## Pathology and histopathology

Length and weight of every euthanized fish was recorded and the condition index was calculated (100*weight/lenght$^3$) [36, 37]. A complete necropsy was performed. Macroscopic changes in the inner organs were evaluated. Kidneys were removed and cut longitudinally. One part was immediately fixed in 10% buffered formalin for histopathological examination. The other half was fixed in RNAlater® (DNA stabilisation solution, Insel Gruppe AG, Switzerland) for qPCR analysis. At the Centre for Fish and Wildlife Health (CFWH), formalin fixed samples were routinely prepared for histology and stained with haematoxylin and eosin (HE). HE stained slides were examined by light microscopy (Nikon Eclipse E400, Ryf AG, Switzerland). In renal histopathology, the following criteria were considered as acute changes: proliferation of interstitial tissue, vasculitis with thrombi, interstitial haemorrhage, vessel wall necrosis and inflammation with mainly macrophages and lymphocytes, interstitial necrosis, interstitial inflammation predominantly with macrophages. Chronic changes were characterized by interstitial inflammation predominantly with macrophages, interstitial fibrosis, tubulonephrosis and tubuloneogenesis [31]. These histopathological changes were graded as 0 (no), 1 (scattered), 2 (mild), 3 (mild to moderate), 4 (moderate), 5 (moderate to severe) to 6 (severe) according to Bettge et al. (2009) [31]. Presence of *T. bryosalmonae* was examined on one histological section through the entire kidney. Six randomly selected high power fields (HPF) (magnification: 400x) were evaluated and parasite abundance was classified ranging from 0 (no parasites / 6 HPFs), 1 (1–5 parasites / 6 HPFs), 2 (5–10 parasites / 6 HPFs)), 3 (10–20 parasites / 6 HPFs), 4 (20–50 parasites / 6 HPFs), 5 (50–100 parasites / 6 HPFs) to 6 (> 100 parasites in renal haematopoietic tissue, vessels and / or tubules / 6 HPFs) [31].

## qPCR for detection of *T. bryosalmonae* DNA in kidney tissue, conventional PCR and sequencing

Fish negative or suspicious for *T. bryosalmonae* by histology were screened for traces of parasite DNA by qPCR. Before extraction, kidneys were homogenized in a 2 ml tube containing 0.2 ml ATL buffer (QIAGEN, Germany) with a 5 mm diameter steel bead (QIAGEN, Germany) using a tissue lyser (QIAGEN, Germany) with a shaking frequency set at 30 shakes per sec for 2 min. Genomic DNA was extracted by using a DNeasy Blood & Tissue Kit (QIAGEN, Germany) according to the manufacturer's instructions. DNA was finally eluted in 100 μl of AE buffer (QIAGEN, Germany) and stored at -20˚C. Tissue from a known positive brown trout was included as extraction control. qPCR was performed targeting *T. bryosalmonae* 18s rDNA (Acc. N.: AF190669) using a TaqMan method according to Bettge et al. (2009a) [38]. All reactions were carried out in duplicates. In each qPCR run, negative controls and positive controls were included in duplicates. Non-target controls (DNAse free water) within the qPCR never showed amplification, while the positive controls (sample known positive for *T. bryosalmonae*) were always amplified, showing no qPCR inhibition.

To confirm the specificity of selected qPCR results, selected samples showing a positive qPCR result were re-evaluated by conventional PCR according to Morris et al. (2002) [39] with some modifications. Primer pairs described by Kent et al. (1998) [40] were used. PCR products were purified with WIZAR RD®SV Gel and PCR Clean-Up System (Promega AG,

Switzerland). The products were checked on a 1,5% agarose gel for amplification and molecular weight and sent for sequencing *(*Microsynth AG, Switzerland). Sequencing results were determined by BLAST-n based on a search in the GenBank database (www.ncbi.nlm.nih.gov).

## DNA extraction from filters and qPCR for *T. bryosalmonae* and *F. sultana*

DNA extraction from the filters was performed according to Miya et al. (2016) [41] with the DNeasy Blood &Tissue Kit (QIAGEN, Germany), leaving the housing intact. qPCR targeting *T. bryosalmonae* 18s rDNA was carried out in duplicates as described above. DNAse free water served as negative control. As internal control, Exo IPC Mix and IC DNA (TaqMan University MMix w Exog IntPostC, Applied Biosystems, MA, USA) was used [26]. qPCR targeting *Fredericella sp*. 16s DNA was performed in duplicates according to Carraro et al. (2017) [42]. DNAse free water served as negative control, extracted DNA from fresh *F. sultana* cultures as positive control.

## Monitoring of migration by tagging wild brown trout

Trout were sampled by electrofishing in the river Wutach at two stretches of 280 m each, on the 24th of August 2017. The first stretch (start 2'677'061.2/1'289'694.7, end 2'677'155.2/ 1'289'950.7) was situated 1200 m downstream of the estuary of the Ehrenbach, and the second stretch (start 2'677'814.2/1'291'059.7, end 2'678'012.2/1'291'261.7) was situated 280 m upstream of the estuary. Those two sites were selected to monitor fish possibly migrating into the nearby tributary. Accessibility of the river was an additional criterion. In total 162 brown trout and two rainbow trout were captured, 68 brown trout at the upper location, 96 trout (94 brown trout, 2 rainbow trout) at the downstream location.

Fish were anaesthetized with 1 ml clove oil (80–95% Eugenol®, Merck, Switzerland) in aerated 30 l water, the body cavity was opened by a 3–4 mm incision and a half-duplex (HDX) PIT-tag (OREGON RFID, OR, USA) was implanted in every fish for individual recognition. Depending on the size of the fish, a 12 mm (fish < 150 mm) or 23 mm PIT-tag (fish > 150 mm) was implanted. After implantation, fish were kept for at least 30 min in an aerated tank (100x50x30 cm) for recovery. Afterwards, trout were released into the respective river stretch.

For detection of migrating fish, a single HDX-antenna with an automatic registration and single reader (OREGON RFID, OR, USA) was installed 300 m upstream of the estuary of Ehrenbach on August 24th 2017. It was located approximately 15 m upstream from the site of the removed barrier (2'677'531.8/1'291'207.6). The setup registered the PIT-ID and time for each tagged fish in the electromagnetic field of the antenna, but only allowed for presence-absence detection. A multiplex array was not feasible, as there was electromagnetic interference from the environment. Due to heavy flooding and flotsam, the antenna design had to be a swim-over antenna. It consisted of a double coil spanning the entire cross section of the Ehrenbach and was fixed on the river bed. The reading range was approximately 15–35 cm above the bottom for the 23 mm PIT-tags and almost covered the entire water column. For the smaller 12 mm PIT-tags the reading range was around 15 cm or less above the bottom, resulting in a lower detection probability. The setup was kept running until January 6th 2020. It was out of order due to software issues, power failure or damage by flooding, from October 7th until November 15th 2017, December 3rd until December 4th 2018, January 4th until March 8th 2018, April 27th until July 4th 2018 and October 16th until October 22nd 2018.

## Water temperature

Water temperature from Ehrenbach (2'674'615/1'292'318) and Wutach (2'676'398/1'289'323) was recorded between 2013 and 2018, from April to October every 2 hours by Onset TidbiT v2

Temp Loggers (location of loggers, see Fig 1). For 2019, no data were available from Ehrenbach, because the logger was lost by vandalism. Mean values for each day were calculated and plotted.

## Statistics

Migration data, fish data and temperature data were analysed using *R* 3.1.2 (R Core Team 2018) as well as Excel 2016 [43]. Condition index was calculated with the following formula $100^*weight/lenght^3$. Mean values and standard deviation of length, weight, condition index, infection severity and parasite abundance values were calculated per group using Excel 2016. Mean values of temperature data per day were calculated. Afterwards, temperature values over a period of one year were plotted using Excel 2016. Migration data were analysed using a self-made algorithm in R.

## Results

### Fish parameters and PKD status

Length, weight and condition index of brown trout sampled from Wutach and Ehrenbach between 2013 and 2019 are shown in Table 1. Over the years, condition index varied within normal limits for brown trout, according to the criteria of Froese et al. (2016) [36].

At the first sampling in 2013, before removal of the migration barrier, 15 brown trout sampled in the Ehrenbach showed no signs of PKD, neither macroscopically nor histologically. By qPCR, no *T. bryosalmonae* DNA was detected. In the Wutach, 9 out of 11 animals (81.8%) showed histological signs of an infection with *T. bryosalmonae*. Eight animals showed acute lesions with vascular thrombosis, interstitial necrosis and haemorrhage, interstitial infiltration with lymphocytes and macrophages and low to high numbers of parasites. One animal showed chronic active changes with interstitial fibrosis and concurrent acute signs as described above. Severity of changes and parasite abundance are summarized in Table 1.

In 2014, the migration barrier was removed. Annual samplings of brown trout YOY were implemented from 2016 to 2019 in the rivers Wutach and Ehrenbach. In brown trout sampled from the Ehrenbach, in none of the examined fish at any time, neither macroscopic nor

**Table 1. Number (n) of fish investigated, length, weight, condition index, PKD prevalence, parasite abundance and associated pathology of *T. bryosalmonae* infection in brown trout in Ehrenbach and Wutach in 2013 (before removal of barrier) and from 2016 to 2019 (after removal of barrier in 2014).**

| | Ehrenbach | | | | | Wutach | | | | |
|---|---|---|---|---|---|---|---|---|---|---|
| | 2013 | 2016 | 2017 | 2018 | 2019 | 2013 | 2016 | 2017 | 2018 | 2019 |
| n | 15 | 27 | 30 | 20 | 20 | 11 | 22 | 26 | 12 | 20 |
| Length (mm), mean±SD | 83±15 | 99±19.2 | 99±8.67 | 99±15.31 | 90±14.14 | 102±9.6 | 102±13 | 101±15.04 | 129±28.34 | 98±12.64 |
| Weight (g), mean±SD | nd | 12±8 | 12±7.23 | 11.2±5.28 | 7.8±2.42 | nd | 11.8±4.38 | 11.8±5.12 | 25±18.73 | 10.5±3.94 |
| Condition index, mean±SD | nd | 1.19±0.14 | 1.19±0.14 | 1.13±0.32 | 1.03±0.11 | nd | 1.09±0.12 | 1.09±0.12 | 1.01±0.09 | 1.09±0.2 |
| Prevalence by histology (%) | 0 | 0 | 0 | 0 | 0 | 82 | 77 | 75 | 92 | 50 |
| Prevalence by qPCR (%) | 0 | 4* | 0 | 0 | 0 | nd | 100 | 100 | 100 | 100 |
| Parasite abundance, mean±SD | 0 | 0 | 0 | 0 | 0 | 2.8±1.3 | 2.1±1.5 | 2.7±1.5 | 2.8±1.5 | 1.9±1.4 |
| Pathology severity, mean±SD | 0 | 0 | 0 | 0 | 0 | 3.6±1.9 | 3.0±1.8 | 3.6±1.6 | 3.5±1.5 | 2.1±1.5 |

*one positive animal by qPCR.

Shown are mean values ± standard deviation (SD) of length, weight, condition index, parasite abundance and pathology severity grades. PKD prevalence was calculated by number of positive fish divided by total number of examined fish by histology (Prevalence by histology) or examined by qPCR (Prevalence by qPCR). Parasite abundance and pathology severity were examined by histology. Parasite abundance was classified ranging from 0 (no parasites / 6 HPFs) to 6 (> 100 parasites in renal hematopoietic tissue, vessels and / or tubules / 6 HPFs). Pathology severity was graded from 0 (no changes) to 6 (severe changes); nd = not done.

histological signs of an infection with *T. bryosalmonae* were found. All animals from the Ehrenbach were additionally investigated by qPCR. All examined brown trout revealed to be negative by qPCR, with the exception of one animal in 2016. This single positive result was confirmed by sequencing. The product showed a 100% identity to the *T. bryosalmonae* 18S ribosomal RNA gene (Accession n° EU570235.1). All other samples tested by qPCR were negative (Table 1).

From 2016 to 2019, 50 to 92% of investigated brown trout from the river Wutach showed histological signs of PKD with acute lesions and interstitial extrasporogonic parasite stages (Table 1). Several specimens showed intratubular spores. By qPCR, all remaining brown trout were positive for *T. bryosalmonae* DNA with a total prevalence of 100% during the investigation period (Table 1).

## qPCR for detection of *F.sultana* and *T. bryosalmonae* by eDNA

qPCR showed negative results for *Fredericella* sp. and *T. bryosalmonae* in all three field replicates sampled in the Ehrenbach. The six water samples collected at the two locations in the Wutach (three replicates at two locations) showed a positive result for *Fredericella* sp. in all samples beside one replicate at the upstream location. qPCR for *T. bryosalmonae* was positive in three replicates, two replicates at the downstream location and one replicate at the upstream location. All other replicates did not detect parasite DNA in the water.

## Fish migration

In total, 162 brown trout and two rainbow trout from the Wutach were PIT-tagged. Length, weight and condition index of tagged brown trout are shown in Table 2.

On 5th of September 2017, the first brown trout migrating from the Wutach into the Ehrenbach was registered. Monitoring was continued until 6th of January 2020. Migration of brown trout was exclusively detected between the end of September and the beginning of March of the respective years (one rainbow trout was detected in March and April 2018). During the investigation period, 19 tagged brown trout and one rainbow trout migrated from the Wutach into the Ehrenbach (12% of all tagged trout) (Fig 2). 143 brown trout and one rainbow trout were never detected (Fig 2). No tagged fish below 150 mm total length were detected in the Ehrenbach. Three fish were detected in consecutive years, one of these individuals even during every spawning season of the investigation period. The other two brown trout were detected in two consecutive years.

## Water temperature

In 2013, water temperature was higher in the Wutach compared to the Ehrenbach. However, in the Ehrenbach, water temperature was increasing during the investigation period, levelling similar to the temperature measured in the Wutach (Fig 3). In the Ehrenbach, water temperature exceeded 15°C over 8 non-consecutive days in 2013 (3 days in July and 5 day in August). In 2017, the time span increased to 65 non-consecutive days (July till August with single days below 15°C). In 2018, temperature surpassed 15°C on 51 non-consecutive days distributed

**Table 2. Length, weight and condition index of PIT-tagged brown trout according to sampling site.**

| sampling site | N | total length in mm | | | | weight in g | | | | condition index | | | |
|---|---|---|---|---|---|---|---|---|---|---|---|---|---|
| | | min. | max. | mean | median | min. | max. | mean | median | min. | max. | mean | median |
| upstream of Ehrenbach | 68 | 90 | 432 | 247 | 251 | 9 | 941 | 251 | 194 | 0.85 | 1.78 | 1.21 | 1.19 |
| downstream of Ehrenbach | 94 | 102 | 471 | 288 | 295 | 12 | 1072 | 329 | 296 | 1.03 | 1.47 | 1.18 | 1.17 |

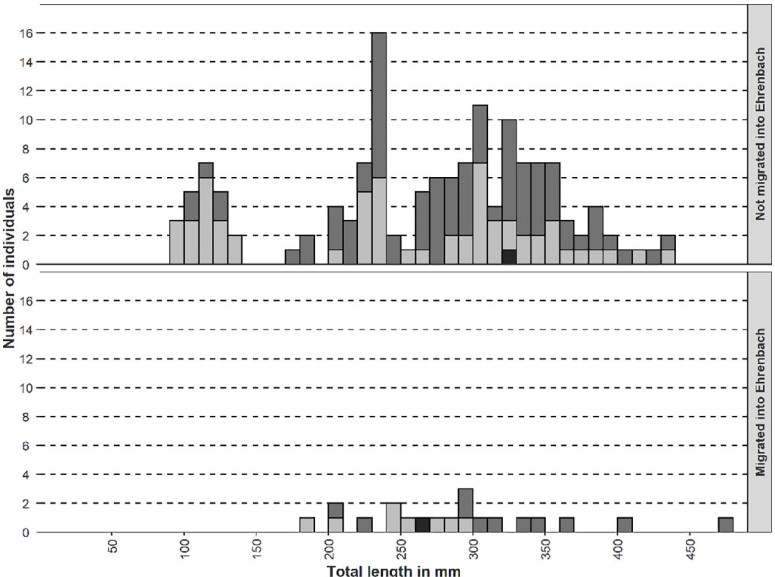

**Fig 2. Length frequency distribution of trout migrating into the Ehrenbach.** The total length refers to the length at the moment of tagging. Light grey bars = brown trout tagged in upstream stretch of the Wutach, grey bars = brown trout tagged in downstream stretch of the Wutach, dark grey bar = rainbow trout.

over July and August, again with single cooler days in between. In the Wutach, water temperature exceeded 15°C over 48 non-consecutive days in 2013 (July till August), over 83 non-consecutive days in 2017 (June till August) and over 80 non-consecutive days in 2018 (June till August). Similar to the Ehrenbach, temperature showed some fluctuations with single cooler days in this warm time period. As time span of temperature <15°C was limited to single days in the mentioned periods and temperature fluctuated around 15°C, non-consecutive days are recorded.

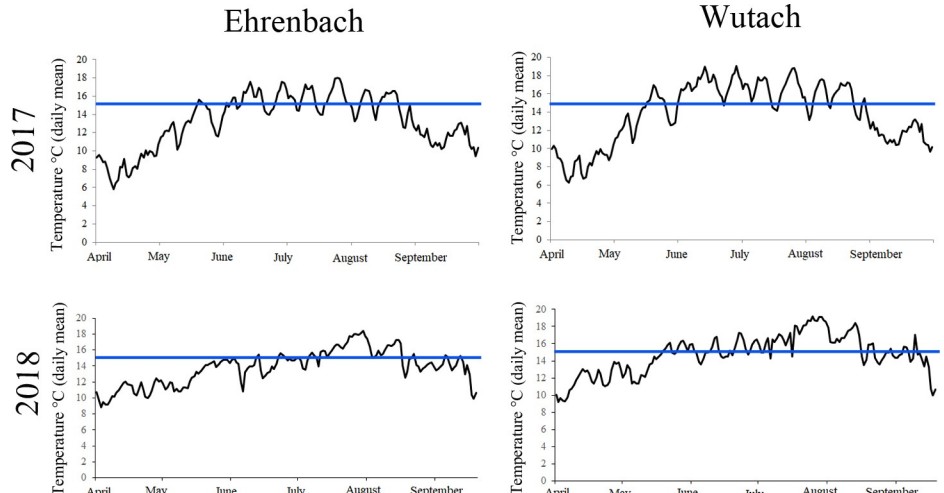

**Fig 3. Temperature profiles from the rivers Ehrenbach and Wutach, 2017 and 2018.** PKD is temperature dependent with the disease getting clinically relevant at water temperature above 15°C. The blue line in the temperature profiles marks the 15°C line.

## Discussion

River regulation for flood protection purposes and the construction of barriers for hydropower generation or agricultural irrigation has led to an increased number of artificially impacted water bodies. Migrating fish populations are strongly affected by intra- and inter-stream barriers that cause river network fragmentation, constraining productivity or preventing migration for spawning. However, the problem is well recognized and several restoration projects are ongoing in Switzerland and Germany (https://plattform-renaturierung.ch/). For brown trout, which migrate considerable distances, free connectivity of river systems is essential to reach spawning habitats. In addition, increased connectivity also increases biodiversity facilitating migration also for smaller fish species [44] and invertebrates. On the other hand, spreading of infectious agents possibly detrimental for vulnerable fish populations can be enhanced by migration. Spreading of pathogens by migration has been shown for different animal groups including mammals, birds, fish, and insects [45, 46]. However, migration also allows hosts to escape from infected habitats, reduces disease levels when infected animals do not migrate successfully, and may lead to the evolution of less-virulent pathogens [47].

PKD is one example of an emerging fish disease rapidly spreading throughout Europe during the last decades [11, 13, 48]. This spreading is due to a complex interaction of multiple factors [13, 26, 31, 49–51]. It has been shown that distribution of PKD is related to the occurrence of bryozoan, mainly *F. sultana* [28, 42]. In our eDNA investigation, no *Fredericella* sp. DNA was detectable in water from the Ehrenbach. In contrast, in the Wutach, both *Fredericella* sp. and *T. bryosalmonae* DNA was detectable. However, local inspections along the Ehrenbach have revealed presence of *Plumatella sp.* (Hartikainen, pers. comm.), a bryozoan which, beside *F. sultana*, may also act as a host for *T. bryosalmonae* [52]. Therefore, even if *Fredericella* occurrence was not confirmed in the tributary, a possible final host to complete the parasite cycle seems to be present. Distribution of freshwater bryozoan is dependent on elevation above sea level, water temperature, water velocity, and eutrophication [53]. The temperature profile of the rivers Wutach and Ehrenbach showed that water temperature was higher in the PKD positive Wutach. The growth of *F. sultana* colonies increases when water temperature reaches 8 to 12˚C in spring. Also in the Ehrenbach, water temperature was mostly > 8˚C during the whole year. Therefore, in the Ehrenbach, water temperature alone is not sufficient to explain host and parasite distribution. Further investigations are needed to confirm bryozoan occurrence in the Ehrenbach as well as the importance of different bryozoan species in the parasite cycle.

Brown trout migration from the Wutach into the Ehrenbach was mainly seen in winter months, from December to April. This supports the hypothesis that the Ehrenbach is an important spawning stream for brown trout. On the other hand, PKD is a disease which gets clinically apparent primarily in YOY, an age class not migrating upstream for spawning. However, excretion of spores has been shown in adult brown trout, up to 5 years post exposure [30]. Therefore, transport and excretion of infectious spores by adult migrating brown trout cannot be excluded. Furthermore, bryozoans have been shown to shed infective spores all year round and brown trout get infected. When infected fish are transferred to higher temperature, they can develop clinical disease [34]. Therefore, even migrating adult trout could transmit the parasite during winter months and complete the cycle. In the first year after barrier removal, qPCR showed *T. bryosalmonae* DNA in a single fish. This was a unique, isolated finding over the whole survey period, not associated with disease, tissue changes or parasite detection by histology. Whether or not this detection of *T. bryosalmonae* DNA was linked to migrating trout remains unresolved.

Considering the technical malfunction of the PIT antenna over cumulated 91 days, between October 2017 and March 2018 (i. e. mainly during the spawning migration period) due to

flooding and electrical supply failure, the number of registered events might not reflect the accurate number of all migration movements. Nevertheless, the total detection rate over 27 months of tagged brown and rainbow trout migrating into the Ehrenbach reached 12% (20 out of 164). Only adult fish were detected. None of the 21 tagged YOY fish were registered in the Ehrenbach. The Ehrenbach was probably used by adult brown trout from the Wutach as a spawning tributary. Bagliniere & Maisse (2002) confirmed in a comprehensive study that spawning of brown trout took place mainly in the tributaries of rivers, but also in the main river [54]. On the other hand, the detection range of the RFID-antenna was suboptimal for smaller 12 mm PIT-tags, compared to the larger 23 mm PIT-tags. Therefore, smaller fish (< 150 mm total length) were less likely to be registered. Furthermore, the mortality of small fish can be assumed to be naturally higher than of larger fish, which leads to a lower chance of detection. In addition, it can be assumed that brown trout affected by *T. bryosalmonae* are less mobile. Therefore, recognition of brown trout migrating from the Wutach into the Ehrenbach might be biased, detecting only healthy fish.

## Conclusion

Our study illustrates that, after removal of an impassable barrier, adult brown trout were migrating upstream during winter months. This supports the importance of continuity of this stream for habitat maintenance and productivity. Although migration of brown trout from a PKD-positive river into a PKD-negative tributary was confirmed, upstream spreading of the disease was not observed over a five-year period. Therefore, the benefit of restoration of the river corridor seems to exceed the risk of transmission of infectious agents at least over limited periods in the study area. However, findings are based on a single river system. This highlights the importance of further monitoring studies in future restoration projects, with a focus on possible parasite spread.

## Acknowledgments

We thank the fisheries association „Oberes Wutachtal Stühlingen e.V."for help during electro-fishing, the staff of the histology laboratory of the Institute of Animal Pathology for preparation of the histological sections and Christopher Robinson for evaluating data of the temperature loggers. In addition, we are grateful to James Ord for the correction of the English language.

## Author Contributions

**Conceptualization:** Heike Schmidt-Posthaus.

**Data curation:** Heike Schmidt-Posthaus, Armin Peter.

**Funding acquisition:** Heike Schmidt-Posthaus.

**Investigation:** Heike Schmidt-Posthaus, Ernst Schneider, Nils Schölzel, Regula Hirschi, Armin Peter.

**Methodology:** Moritz Stelzer.

**Project administration:** Heike Schmidt-Posthaus.

**Writing – original draft:** Heike Schmidt-Posthaus, Armin Peter.

**Writing – review & editing:** Heike Schmidt-Posthaus, Ernst Schneider, Nils Schölzel, Regula Hirschi, Moritz Stelzer, Armin Peter.

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
