## [Decision Letter · Decision Letter 0]

5 Aug 2020

PONE-D-20-18486

The role of migration barriers for dispersion of Proliferative Kidney Disease – balance between disease emergence and habitat connectivity

PLOS ONE

Dear Dr. Schmidt-Posthaus,

Thank you for submitting your manuscript to PLOS ONE. After careful consideration, we feel that it has merit but does not fully meet PLOS ONE’s publication criteria as it currently stands. Therefore, we invite you to submit a revised version of the manuscript that addresses the points raised during the review process.

ACADEMIC EDITOR: 

Dear Authors of the manuscript Number PONE-D-20-18486 entitled "The role of migration barriers for dispersion of Proliferative Kidney Disease – balance between disease emergence and habitat connectivity". I read myself this interesting manuscript and I obtained comments from three eminent experts in Malacosporean parasite and PKD disease. Despite the divergent opinions from these three reviewers, I think the manuscript has still merit for being published in Plos ONE. As expressed from the three reviewers, there are some main concerns that involve the possibility to make inference about the results obtained. This is especially true if looking at the fact that the study was 1) limited in the sample size of fish host (brown trout) analysed and 2) lacking of relevant information concerning the environmental features that characterize the study area and 3) most importantly concerning the occurrence of freshwater bryozoan hosts (mostly *Fredericella sultana*) in both the areas investigated. The latter point is essential to understand the results here obtained that suggest how connectivity in the fish host is not affecting the introduction of the parasite in the area that was free from the parasite before the reopening of migratory barriers in 2014. In fact, the lack of bryozoan hosts in the area reopened to fish connectivity might be the reason for such a pattern considering that the availability for a bryozoan host is a determining factor in the ability of the parasite *Tetracapsuloides bryosalmona*e to complete the life cycle and being carried around by the intermediate (fish) host. The manuscript is also limited to a very narrow area and the authors should have better described the features of the area here studied and express this limit in the discussion. As a minor consideration, I would recommend a better figure (in terms of quality) of the studied area because it is essential for the main aim of this manuscript.

I`d like to open up to the possibility to publish this manuscript if the authors can cover these gaps, especially providing proof that both areas here investigated show viable freshwater bryozoan populations that can sustain the interpretation of their results. Therefore, I`m going to suggest "major revision".

We look forward to receiving your revised manuscript.

Kind regards,

Paolo Ruggeri, PhD

Academic Editor

PLOS ONE

Journal Requirements:

2. Please include a caption for figure 4.

3.We note that [Figure(s) 1] in your submission contain [map/satellite] images which may be copyrighted. All PLOS content is published under the Creative Commons Attribution License (CC BY 4.0), which means that the manuscript, images, and Supporting Information files will be freely available online, and any third party is permitted to access, download, copy, distribute, and use these materials in any way, even commercially, with proper attribution. For these reasons, we cannot publish previously copyrighted maps or satellite images created using proprietary data, such as Google software (Google Maps, Street View, and Earth). For more information, see our copyright guidelines: http://journals.plos.org/plosone/s/licenses-and-copyright.

1.    You may seek permission from the original copyright holder of Figure(s) [1] to publish the content specifically under the CC BY 4.0 license. 

4. Please include a copy of Table 3 which you refer to in your text on page 11.

5. Please amend your list of authors on the manuscript to ensure that each author is linked to an affiliation. Authors’ affiliations should reflect the institution where the work was done (if authors moved subsequently, you can also list the new affiliation stating “current affiliation:….” as necessary).

6.Thank you for stating the following in the Financial Disclosure section:

[The study was financially supported by the Federal Office for the Environment (FOEN), Bern, Switzerland (HS, AP) (www.bafu.admin.ch). The funders had no role in study design, data collection and analysis, decision to publish, or preparation of the manuscript.].   

We note that one or more of the authors are employed by a commercial company: FishConsulting GmbH.

Please respond by return email with an updated Funding Statement and Competing Interests Statement and we will change the online submission form on your behalf.

Reviewers' comments:

Reviewer's Responses to Questions

**Comments to the Author**

1. Is the manuscript technically sound, and do the data support the conclusions?

Reviewer #1: No

Reviewer #2: Partly

Reviewer #3: Partly

2. Has the statistical analysis been performed appropriately and rigorously? 

Reviewer #1: No

Reviewer #2: No

Reviewer #3: Yes

3. Have the authors made all data underlying the findings in their manuscript fully available?

Reviewer #1: No

Reviewer #2: No

Reviewer #3: Yes

4. Is the manuscript presented in an intelligible fashion and written in standard English?

Reviewer #1: No

Reviewer #2: No

Reviewer #3: Yes

5. Review Comments to the Author

Reviewer #1: I have several major conceptual criticisms of this work, which I will expand upon below. I understand that word limits often restrict methods details, but the methods, as they are described in the submission, are not detailed enough to be repeated. In addition, the manuscript does not seem like it is ready for submission, as there are several formatting anomalies.

The methods of the study do not address the objective as it is outlined in the introduction: “The aim of the study was to investigate the role of a migration barrier in limitation of pathogen transport by migrating fish…”. These data are not generalizable to other diseases and the study design is incapable of addressing this objective. As is pointed out in the manuscript, PKD generally manifests in naïve fish, which are generally YOY. As is also pointed out in the manuscript, YOY brown trout generally do not migrate into tributary streams from larger bodies of water. So the study’s conclusion that “the benefit of restoration of the river corridor seems to exceed the risk of transmission of infectious agents” is based on an extremely biased data set because the fish that are migrating are the same ones that are least likely to succumb to, be susceptible to, or to release parasite spores due to PKD.

Another major flaw is that the study completely ignores the primary bryozoan hosts of T. bryosalmonae. There was no mention of any effort to confirm the presence of bryozoans in the Ehrenbach, which would be necessary for the complete life cycle of T. bryosalmonae, because spores released from infected fish cannot, as far as we know, ever infect other fish. If bryozoans are not present in the Ehrenbach, might the results have been dramatically different if bryozoans were there? On a similar note, the absence of parasite in fish is not necessarily an accurate indicator for the absence of the parasite in bryozoans in the same location. This disease is extremely complex and there are several variables which might decouple infection in the two hosts.

If I am reading the methods correctly, only fish from the Wutach were pit-tagged, which means that there is no movement data for fish that were above the barrier before its removal. These fish may have very different patterns of movement across the two water bodies. This missing information represents a large gap for any epidemiological model of this system.

Reviewer #2: This paper aims to assess whether removal of migration barriers to fish in rivers results in upstream movement of fish and whether their associated parasites also disperse with the fish. This aim is communicated clearly. The research focuses on a relatively rare opportunity to track parasite dispersal in a real river system, centering on a removal of a migration barrier in one river. The paper presents novel results from a field observational experiment. Prevalence of infection was monitored upstream and downstream of a migration barrier one year before the barrier removal, followed by monitoring in five years after the removal of the barrier.

The premise of the study is of applied importance for many river restoration efforts. The research rationale and methodology is in principle presented well, however, not enough details of the study system are given to understand how the parasite might disperse to the upstream reaches of the river, and at what time scales that might be expected to occur. Regardless, the study design and methodology appear sound.

The main concern is that the results are based on a single site. However, it is clear that such investigations are exceedingly difficult to replicate in wild. Regardless it is very important to highlight this issue in the discussion and perhaps recommend ways to improve monitoring in future restoration projects, and highlight the focus on parasite spread. I unfortunately was not able to find Table 3 and the line numbering disappeared around the discussion start, making in difficult to highlight specific issues. I hope the comments will nevertheless be possible to be tracked by the authors.

The writing is clear, and the authors have done a very competent job of the scientific discussion and arguments, BUT editing of English language is needed throughout to improve sentence structure in particular. Otherwise the article is clearly written and logically organised, it is relatively easy to follow. Appropriate controls and confirmatory techniques are taken in most parts. I was not able to correct all the grammar issues as that would have required a substantial amount of work, but have provided a .doc document where I have highlighted particularly noticeable problems in yellow highlight.

If the authors address the suggestions made below, in my opinion this manuscript appears suitable for publication in PLoSOne.

Introduction

- Line 45: why “developing”? Suggest removing “Nowadays”

- Line 47: fragment sentence starting on this line “Thus,…” – please revise.

- Line 49-52: punctuation missing, sentence is really long, and hard to follow due to odd grammar.

- Line 55: “…several current projects” suggests some specific projects are implied (also indicated by the pers. com reference). Suggest revising this sentence to be more general and providing a reference to primary literature.

- Line 59: clearly transmission is important for parasites, but here its combined with the dispersal of the parasite (“transport”). Transmission and dispersal are very different concepts and the sentence should be re-written to distinguish this better.

- Line 61-62: replace “was classified as” with “is”. Classified sounds too formal in this context.

- Line 69: Please review most recent terminology – actinospores and myxospores refer to mxosporeans, suggest using malacospores and fish malacospores instead.

- Does fish to fish transmission occur? Please provide information here e.g. Line 71.

- Line 79: please articulate the sentence above as a question or change wording here. It is not clear what the specific, testable idea (hypothesis) is here. What is being tested? Whether infected fish can move upstream? Whether the parasite is transmitting in the upstream section? Whether the parasite spreads to the new upstream area? It should be pointed out here that migration of adult fish was monitored and disease status in the upstream and downstream YOY populations was also monitored. Please revise this section to be explicit about what the experiment really tests.

- Line 81: It is difficult to understand the extent of this barrier without further information. Please describe what this barrier is like, does it prevent up and downstream movement of fish? What is it exactly? Why is it being removed? What was involved in its removal?

- Finally, it is confusing that the disease manifests in the YOY but these are not migrating. Please provide more information regarding what is known about migration habits of different age classes and what the expectations are relative to the spread and disease in fish populations. Are the migrating adult fish competent hosts and transmit the parasite to other hosts? Please provide references.

- In general the introduction needs a thorough checking of English to improve sentence structure. There are many incomplete sentences and incorrect wordings, which make it tough to follow.

Methods

- Line 99: “own investigation”: whose? Please cite pers comm. NAME if no other reference is available.

- Line 98: It would be important to know if the fish sampled were YOY or adult fish.

- Line 101: When in 2014?

- Line 112: not clear what the “respectively, “ refers to. Please revise.

- Line 122: please use consistent terminology for qPCR or RT PCR or RT qPCR….

- Line 144-147: Are these the same as in the referenced paper? Consider deleting?

- Line 159: was the infection status of these fish checked? Or the infection status of a peer group captured at the same time? Previous investigations showed that prevalence was high in the YOY fish but was that the case for the older fish too? Is it possible the infected fish more less and do not take part in migration?

- The statistics section is not sufficient. This should describe the analyses conducted to a detail where they could be repeated following the description provided here. What tests were done on what data? Please ensure all analyses presented in results section are covered here.

Results

- Table 1: What does the +/- refer to? Please define.

- Figure 1 should be redrawn – the satellite imaging shot does not show the river clearly and its hard to see the confluence. Also, the red font on busy background is difficult to see. The zoomed in map should be presented in a larger context in an inset figure – e,g, its difficult to understand where the border is and where the site is located.

- Figure 2 is not particularly informative. Could be easily deleted or replaced.

- Where is Table 3?

- Line numbers disappers and it difficult to provide direct comments from here forwards…

- Can the possible invasion be really linked to the migrating trout? Using the methods employed here, it does not seem be possible to make a direct link, although such effects may be implied. Please reword to take account of this ambiguity.

- Reporting of Cq values is relatively meaningless. Would it be possible to provide copy numbers per unit of kidney, this would make comparisons more relevant. Taking means of Cq values across samples is also confusing and should be avoided in the table.

- Is the section starting with “From 2016 to..” referring to Wutach? Please add.

- The results section could be made clearer by combining the fish parameters with the PKD monitoring section as these presumably pertain to the same fish. The migration fish are a different batch of animals.

- Table 2: please make separate column for prevalence.

- “On 15th…” migrating from Wutach?

- Did the fish collected and tagged from Ehrenbach move upstream too? The monitoring station was only 20m upstream – is this considered a migration? Why were fish tagged in the Ehrenbach – is the data reported? Suggest reporting results separately for the fish tagged in Wutach and Ehrenbach. Were the same fish detected in multiple years?

- What is the significance of the non-consecutive days?

Discussion

- The discussion needs a re-write, to improve flow and English. Comparisons to other research in the field is limited to the considerations of PKD, it would be interesting to see further discussion relating this research to other disease systems in aquatic context.

- A section on future research directions in this field would be useful, for example, investigations across the full life-cycle.

- It seems one limitation of this study is that the infection status of the tagged fish was not known, and it is not clear if PKD impacted adult fish migrate normally. This should be discussed further.

- The start of discussion should be improved with more details regarding the e.g. wh is restoration “more significant”. Perhaps this is not the wording the authors mean? Referencing here is completely lacking. E.g. “unambiguous benefits” should be backed up by references.

- Include references that discuss or test the spreading of pathogens in association with restoration of connecitivity

- Discuss further the implications of complex life-cycle of parasite for the likelihood of spreading with migration.

- Is the temperature difference significant? In relation to the triggers of PKD and the parasite development and the development of the invertebrate hosts?

- Reference the lack of spore excretion during winter.

- How does the last sentence of the discussion relate to the results generatated in this experiment? Please tie it in better.

- References are not provided in a consistent format.

Reviewer #3: General Comments

This is a straightforward study with only few, albeit incomplete datasets reported. It describes the migratory movements of salmonids native to Wutach river in Switzerland in the context of PKD infectivity. The manuscript is generally well written, although spelling / grammar could be improved in a few places, as descibed under “Specific Comments” below. Although datasets are not complete due to equipment failure, the study is of a publishable quality, subject to addressing specific points below.

Specific Comments

• Lines 37-39: This sentence needs to be rephrased as it, currently, isn’t very clear.

• Liines 49-54: This is a very long sentence. Please rephrase.

• Lines 69-76: It is not correct to refer to T. bryosalmonae stages as myxopsores and actinospores, it’s a malacosporean not a myxosporean.

• Lines 103-104: Could the authors provide a clear justification for the chosen sampling period (24th Aug – 15th Sept)

• Lines 109-111: The authors need to justify why only a single electrofishing site was chosen on the Ehrenbach tributary and two on the Wurach river. This is important since its migration from the Wutach to the Ehrenbach that is under investigation.

• Could the authors confirm the Ehrenbach tributary-Wutach river connectivity with a better-quality image. Current map is not clear enough.

• Line 112: Define YOY

• Line 117: The authors use “condition factor” here but “condition index” elsewhere

• Line 129: What are the criteria that govern each histological grading score. Please define or refer to a previously published study

• Line 131-132: Also define each poiint in the scoring system 0-6 in the context of parasite numbers. Is it numbers of parasites counted in X number of field of views per tissue section with Y number of sections analysed per fish kidney sample? Please properly define, it is currently not very clear.

• Line 149: What is Exo IPC. Please define.

• Lines 153-157: Can the authors provide more detail justifying the why those sites were chosen for the study.

• Line 162: Figure 2 doesn’t provide real added value to the paper. Simply stating the size of tags in the Methods will suffice.

• Line 169-174: Have the authors implemented the tag reading technology previously? If os, please provide a reference.

• Lines 179-181: What do the authors means in saying the logger was lifted. Why?

• Line 191: Line numbering ends here!

• “(table 3). One animal showed chronic active changes with interstitial fibrosis and concurrent acute signs as described above. Severity of changes and parasite abundance are summarized in table 3”. No Table 3 provided. Presumably, this is a typo and should be Table 2.

• Table 2 and legend need to be clearer and the authors need to describe, in the discussion, why only fish sampled in 2013 are PCR negative yet all other signs point to parasite infection / exposure. Parasite tissue distribution is unlikely to homogenous throughout the kidney and so could account for apparent differences between fixed tissue datasets versus qPCR results? Need to clarify that the “parasite abundance” and “pathology severity” columns originate from histological and parasite abundance datasets (from scores 0-6). Define “nd”.

• “162 brown trout and two rainbow trout were PIT tagged. Length of tagged fish ranged from 90 to 471 mm (mean 270.9, median 282), weight from 9 to 1072 (mean 296.6, median 269.5) g. Condition index ranged from 0. 85 to 1.78 (mean 1.19)”. What were the weights / lengths of fish per sampling site?

• “During the investigation period, 19 tagged brown trout and one rainbow trout were migrating from the Wutach into the Ehrenbach (12%) (Fig. 3)” How does this level compare to similar studies? As it wasn’t possible in this study to get a real idea of the number of tagged fish remaining in the Wutach via tag detection versus fish losses / mortalities, it is important in the discusion to refer to previous datasets.

• So, only mature fish migrate with YOY fish remaining in the Wutach. Yet, the majority of mature fish also remain in the Wutach. Are there other known spawning sites within a few Kms of the study sites? Lack of PKD infectivity in mature migrating fish supports a link between fitness and migration with infected fish less inclined to migrate. This is needs to be mentioned in the discussion.

• 2018 NOT 2918

• “Fig. 3. Temperature profiles from Ehrenbach and Wutach, 2013, 2017 and 2018. Red line marks 15°C, the temperature where PKD revealed to be clinically relevant” This is unclear, please clarify.

• “bryozoan, the final host”: Please correct to “definitive host”

• “Even if the minimum necessary number of days with high water temperature for the clinical outbreak of disease is not known and…………” Please re-structure this very long sentence.

• “……..trout does not play a role”: In what?

• “Considering the technical malfunction of the PIT antenna over cumulated 91 days, between October 2017 and March 2018………….” So, data was only used from one complete spawning season due to technical failure?

• “This proves the importance of continuity…………”Change to “This supports the importance……”

• “We thank the fishermen „Oberes Wutachtal Stühlingen e.V.“ for……” Please correct.

• Figure 3 and 4 axes labels need to be improved re. quality.

6. PLOS authors have the option to publish the peer review history of their article (what does this mean?). If published, this will include your full peer review and any attached files.

Reviewer #1: No

Reviewer #2: **Yes: **Hanna Hartikainen

Reviewer #3: No

---

## [Author Response · Author response to Decision Letter 0]

26 Jan 2021

We would like to thank the reviewers for their valuable comments and corrections. 

In this revision we have incorporated the majority of suggestions made by the reviewers. Further, we have discussed those comments that we were not able to address. 

The reviewers’ comments are copied below. Each suggestion/comment is followed by the changes we have made (in italics).

Concerning comments of Reviewer #1:

I have several major conceptual criticisms of this work, which I will expand upon below. I understand that word limits often restrict methods details, but the methods, as they are described in the submission, are not detailed enough to be repeated. In addition, the manu-script does not seem like it is ready for submission, as there are several formatting anoma-lies.

In the method section, more details were added.

The methods of the study do not address the objective as it is outlined in the introduction: “The aim of the study was to investigate the role of a migration barrier in limitation of path-ogen transport by migrating fish…”. These data are not generalizable to other diseases and the study design is incapable of addressing this objective. 

The aim of the study was edited to “The aim of the study was to investigate the possible consequences of the removal of an artificial migration barrier for the upstream transport of Tetracapsuloides bryosalmonae, the causative agent of Proliferative Kidney Disease (PKD) in brown trout, by migrating fish.»

As is pointed out in the manuscript, PKD generally manifests in naïve fish, which are gen-erally YOY. As is also pointed out in the manuscript, YOY brown trout generally do not mi-grate into tributary streams from larger bodies of water. So the study’s conclusion that “the benefit of restoration of the river corridor seems to exceed the risk of transmission of in-fectious agents” is based on an extremely biased data set because the fish that are migrat-ing are the same ones that are least likely to succumb to, be susceptible to, or to release parasite spores due to PKD.

The reviewer is definitely right that adult fish are unlikely to succumb due to PKD. Howev-er, there are indications that adult brown trout up to 5 years old can release spores infec-tive to bryozoa (Soliman et al. 2018).Therefore, it cannot excluded that adult brown trout can transport infective spores. On the other hand, the argument that PKD could be trans-mitted by migrating brown trout is often used to block revitalization programs, at least in Switzerland.

Another major flaw is that the study completely ignores the primary bryozoan hosts of T. bryosalmonae. There was no mention of any effort to confirm the presence of bryozoans in the Ehrenbach, which would be necessary for the complete life cycle of T. bryosalmo-nae, because spores released from infected fish cannot, as far as we know, ever infect other fish. If bryozoans are not present in the Ehrenbach, might the results have been dramatically different if bryozoans were there? On a similar note, the absence of parasite in fish is not necessarily an accurate indicator for the absence of the parasite in bryozoans in the same location. This disease is extremely complex and there are several variables which might decouple infection in the two hosts.

The reviewer is definitely right. By visual inspection, the bryozoan Plumatella sp. was de-tected in the Ehrenbach in former years. This finding is now mentioned in the discussion. Based on the reviewers comment, we now performed an eDNA investigation to monitor occurrence of Fredericella sultana, the proposed main host in the parasite cycle and T. bryosalmonae DNA in the Ehrenbach and the Wutach. The results are mentioned in the new version of the manuscript.

If I am reading the methods correctly, only fish from the Wutach were pit-tagged, which means that there is no movement data for fish that were above the barrier before its re-moval. These fish may have very different patterns of movement across the two water bodies. This missing information represents a large gap for any epidemiological model of this system.

Fish from the Ehrenbach were not PIT tagged, that is correct. As we wanted to monitor mainly movement from the PKD positive river body to the PKD negative river body, movement of fish from above the barrier was not recorded.

Specific Comments:

Line 58: “…still a point of concerns.” De-pluralize to “concern”.

Changed

Line 68: “primary hosts” is probably a better term for bryozoans than “final hosts”

The term was changed into "definitive host" as proposed by reviewer 3.

Line 69 and throughout: “actinospore” and “myxospore” are terms specifically applied to organisms from the class myxosporea (e.g. M. cerebralis), rather than those of malaco-sporea (e.g. T. bryosalmonae). The best terms that we have for malacosporean spores are along the lines of “fish malacospores”, for those infectious to fish, and “bryozoan malaco-spores” for those infectious to bryozoans, etc.

The terms were changed throughout the manuscript.

Line 76: “with a peak in disease severity and myxospore release in late summer / early autumn (30).” Bailey et al. 2017 seems to be a reference for the disease severity part of the statement, but not for the myxospore release part. Tops et al. 2006 is probably the best citation for the later. Note however that neither study is directly linking seasonality. In both cases temperature, is the independent variable. I suggest changing the language to more accurately reflect that nuance, something like “with peaks in disease severity (Bailey et al. 2017) and myxospore release (Tops et al. 2006) corresponding with high temperatures typical of late summer/ early fall.”

Tops, S., Lockwood, W. and Okamura, B., 2006. Temperature-driven proliferation of Tetracapsuloides bryosalmonae in bryozoan hosts portends salmonid declines. Diseases of Aquatic Organisms, 70(3), pp.227-236.

The sentence was changed partly to the reviewer's suggestions. "and myxospore release" was deleted, therefore, the reference Tops et al., 2006 was not included here.

Line 77: You could cite Gay et al. 2001 or Tops et al. 2006 here.

Gay, M., Okamura, B. and De Kinkelin, P., 2001. Evidence that infectious stages of Tetra-capsula bryosalmonae for rainbow trout Oncorhynchus mykiss are present throughout the year. Diseases of Aquatic Organisms, 46(1), pp.31-40.

Tops et al. 2006 was included here. Gay et al. was referred to in the discussion.

Figure 1: The satellite imagery base map makes this figure very hard to see/interpret. The red and black text are especially difficult to read. I would rather see a simple base map with water bodies and sample locations depicted.

Figure 1 was completely revised, showing now a scheme of the location and two own pic-tures of the area where the barrier was removed.

Line 143: should read “18s rDNA” instead of “18 rDNA”

Changed

Throughout methods: Please describe your control schemes in more detail. Were there any extraction controls? If so, how many, with what frequency, and what material was used? Same for qPCR negative controls please.

As positive controls, tissue from known positive brown trout were used. This material was also used as extraction control. Negative and positive controls were described in the re-vised version of the manuscript.

Figure 2: This image does not seem necessary.

Figure 2 was deleted.

Line 169: Did you multiplex antennae to determine directionality of fish movement across the array? This would be a good detail to know. The language right now suggests that it was a singleplex antenna array.

It was a single antenna. It has been changed in the methods section to make it more clear. A multiplex setup has been tested, but was not feasible due to lots of heavy electromagnet-ic interferences. It would have been nice to know the direction of migrating fish, but this setup could only provide presence-absence information, which was sufficient to answer the research objective, if fish from the river Wutach enter Ehrenbach and are able to pass the new block ramps.

Line184: Are there any specific packages or macros that need to be referenced? Methods are intended to be repeatable, but this “statistics” sub-section only mentions that some software was used. What tests were performed? I also did not see any reference to statis-tical tests in the results or discussion sections.

There were no statistical tests used. The only package worth mentioning which was used is ggplot in R, which is now referenced. The migration data were analysed using a self-made algorithm for filtering and processing the RFID data, which is now also mentioned.

In addition, the section was supplemented with the values calculated.

Line191: Maybe this is more of an editorial issue, but I think this should read “… according to the criteria of Froese et al. 2016 (32).”

Changed

We lost line numbers after line 191.

Line numbering was added.

“In brown trout originating from the Ehrenbach, in none of the examined fish at any time, neither macroscopic nor histological signs of an infection with T. bryosalmonae were found.”

“originating” seems like a poor word to use here. I think you mean “sampled”. Otherwise, what is meant by “originating”?

“originating” was changed to “sampled”

In the “Water Temperature” sub-section the font style seems to change.

The font style was adapted.

“During the cold season excretion of T. bryosalmonae spores by brown trout does not play a role.”

I don’t think that this sentence accurately describes anything in particular. It should be re-vised or removed.

The sentence was deleted.

Throughout the manuscript: All of the text should be reviewed by someone who is well versed in English grammar.

The manuscript was reviewed by a native speaker. His name is mentioned in the Acknowledgements.

Concerning comments of Reviewer #2:

Reviewer #2: This paper aims to assess whether removal of migration barriers to fish in rivers results in upstream movement of fish and whether their associated parasites also disperse with the fish. This aim is communicated clearly. The research focuses on a rela-tively rare opportunity to track parasite dispersal in a real river system, centering on a re-moval of a migration barrier in one river. The paper presents novel results from a field ob-servational experiment. Prevalence of infection was monitored upstream and downstream of a migration barrier one year before the barrier removal, followed by monitoring in five years after the removal of the barrier.

The premise of the study is of applied importance for many river restoration efforts. The research rationale and methodology is in principle presented well, however, not enough details of the study system are given to understand how the parasite might disperse to the upstream reaches of the river, and at what time scales that might be expected to occur. Regardless, the study design and methodology appear sound.

The main concern is that the results are based on a single site. However, it is clear that such investigations are exceedingly difficult to replicate in wild. Regardless it is very im-portant to highlight this issue in the discussion and perhaps recommend ways to improve monitoring in future restoration projects, and highlight the focus on parasite spread. I unfor-tunately was not able to find Table 3 and the line numbering disappeared around the dis-cussion start, making in difficult to highlight specific issues. I hope the comments will nev-ertheless be possible to be tracked by the authors.

The mistakes regarding table 3 and the line numbering were corrected. The authors apolo-gize for this mistake. Furthermore, an outlook was added in the discussion: "However, findings are based on a single river system. This highlights the importance of further moni-toring studies in future restoration projects, with a focus on possible parasite spread."

The writing is clear, and the authors have done a very competent job of the scientific dis-cussion and arguments, BUT editing of English language is needed throughout to improve sentence structure in particular. 

The manuscript was reviewed by a native speaker. His name is mentioned in the Acknowledgements.

Otherwise the article is clearly written and logically organised, it is relatively easy to follow. Appropriate controls and confirmatory techniques are taken in most parts. I was not able to correct all the grammar issues as that would have required a substantial amount of work, but have provided a .doc document where I have highlighted particularly noticeable prob-lems in yellow highlight.

Parts highlighted in yellow were re-written. Furthermore, the manuscript was corrected by a native speaker.

If the authors address the suggestions made below, in my opinion this manuscript appears suitable for publication in PLoSOne.

Introduction

- Line 45: why “developing”? Suggest removing “Nowadays”

The sentences were changed accordingly.

- Line 47: fragment sentence starting on this line “Thus,…” – please revise.

The sentence was changed.

- Line 49-52: punctuation missing, sentence is really long, and hard to follow due to odd grammar.

The sentence was changed.

- Line 55: “…several current projects” suggests some specific projects are implied (also indicated by the pers. com reference). Suggest revising this sentence to be more general and providing a reference to primary literature.

The sentence was changed and a reference was added.

- Line 59: clearly transmission is important for parasites, but here its combined with the dispersal of the parasite (“transport”). Transmission and dispersal are very different con-cepts and the sentence should be re-written to distinguish this better.

The sentence was rewritten.

- Line 61-62: replace “was classified as” with “is”. Classified sounds too formal in this con-text.

The sentence was changed.

- Line 69: Please review most recent terminology – actinospores and myxospores refer to mxosporeans, suggest using malacospores and fish malacospores instead.

The term was changed throughout the manuscript.

- Does fish to fish transmission occur? Please provide information here e.g. Line 71.

No, fish to fish transmission does not occur to the current knowledge. The sentence was re-written.

- Line 79: please articulate the sentence above as a question or change wording here. It is not clear what the specific, testable idea (hypothesis) is here. What is being tested? Whether infected fish can move upstream? Whether the parasite is transmitting in the up-stream section? Whether the parasite spreads to the new upstream area? It should be pointed out here that migration of adult fish was monitored and disease status in the up-stream and downstream YOY populations was also monitored. Please revise this section to be explicit about what the experiment really tests.

The aim of the study and the whole paragraph were re-phrased to describe the experiment in a better way.

- Line 81: It is difficult to understand the extent of this barrier without further information. Please describe what this barrier is like, does it prevent up and downstream movement of fish? What is it exactly? Why is it being removed? What was involved in its removal?

A whole paragraph describing the barrier was added.

- Finally, it is confusing that the disease manifests in the YOY but these are not migrating. Please provide more information regarding what is known about migration habits of differ-ent age classes and what the expectations are relative to the spread and disease in fish populations. Are the migrating adult fish competent hosts and transmit the parasite to other hosts? Please provide references.

These topics were discussed and references were provided.

- In general the introduction needs a thorough checking of English to improve sentence structure. There are many incomplete sentences and incorrect wordings, which make it tough to follow.

The introduction was thoroughly revised.

Methods

- Line 99: “own investigation”: whose? Please cite pers comm. NAME if no other reference is available.

A name was added.

- Line 98: It would be important to know if the fish sampled were YOY or adult fish.

YOY were sampled. This was added.

- Line 101: When in 2014?

The barrier was removed in May to June 2014. A paragraph describing the migration barri-er more in detail was added.

- Line 112: not clear what the “respectively, “ refers to. Please revise.

The sentence was changed.

- Line 122: please use consistent terminology for qPCR or RT PCR or RT qPCR….

The terms were equalized.

- Line 144-147: Are these the same as in the referenced paper? Consider deleting?

The primer information was deleted.

- Line 159: was the infection status of these fish checked? Or the infection status of a peer group captured at the same time? Previous investigations showed that prevalence was high in the YOY fish but was that the case for the older fish too? Is it possible the infected fish more less and do not take part in migration?

As fish usually have to be killed to check for the PKD status, animals included in the tag-ging experiment could not be previously checked for infection with T. bryosalmonae. How-ever, as they were living in the same river part as the YOY brown trout examined, pres-ence of the parasite in the environment and possibility for infection can be expected.

- The statistics section is not sufficient. This should describe the analyses conducted to a detail where they could be repeated following the description provided here. What tests were done on what data? Please ensure all analyses presented in results section are cov-ered here.

There were no specific tests performed on the data. The software information was added in the manuscript.

Results

- Table 1: What does the +/- refer to? Please define.

The table legend was changed.

- Figure 1 should be redrawn – the satellite imaging shot does not show the river clearly and its hard to see the confluence. Also, the red font on busy background is difficult to see. The zoomed in map should be presented in a larger context in an inset figure – e,g, its dif-ficult to understand where the border is and where the site is located.

Figure 1 was completely revised, showing now a scheme of the location and two own pic-tures of the area where the barrier was removed.

- Figure 2 is not particularly informative. Could be easily deleted or replaced.

Figure 2 was deleted.

- Where is Table 3?

This was a mistake. Table 2 shows the data. This was corrected throughout the manu-script.

- Line numbers disappears and it difficult to provide direct comments from here forwards…

Line numbers were added.

- Can the possible invasion be really linked to the migrating trout? Using the methods em-ployed here, it does not seem be possible to make a direct link, although such effects may be implied. Please reword to take account of this ambiguity.

A few sentences were added in the Discussion section. 

- Reporting of Cq values is relatively meaningless. Would it be possible to provide copy numbers per unit of kidney, this would make comparisons more relevant. Taking means of Cq values across samples is also confusing and should be avoided in the table.

The Cq values were deleted in the results and in table 2. qPCR was performed to evaluate presence or absence of T. bryosalmonae DNA. Therefore, we refrained to examine copy number per unit of kidney. In the present manuscript version we just refer to presence or absence of parasite DNA. 

- Is the section starting with “From 2016 to..” referring to Wutach? Please add.

The information was added.

- The results section could be made clearer by combining the fish parameters with the PKD monitoring section as these presumably pertain to the same fish. The migration fish are a different batch of animals.

The two sections were combined as well as table 1 and 2.

- Table 2: please make separate column for prevalence.

A separate column was included.

- “On 15th…” migrating from Wutach?

The sentence was adapted accordingly.

- Did the fish collected and tagged from Ehrenbach move upstream too? The monitoring station was only 20m upstream – is this considered a migration? Why were fish tagged in the Ehrenbach – is the data reported? 

In the Ehrenbach, fish were not tagged.

Suggest reporting results separately for the fish tagged in Wutach and Ehrenbach. Were the same fish detected in multiple years?

There were no fish tagged in Ehrenbach, all tagged fish were caught in the river Wutach. 

There were 3 fish that were detected in the Ehrenbach in multiple years. Two of them were detected in two consecutive years and one was detected in 2017, 2018, 2019 and on the last day of operation in 2020.

- What is the significance of the non-consecutive days?

Temperature showed some fluctuation in the warm period, with some cooler days spread-ed inbetween. However, as time span of cooler periods were restricted to single days and temperature stayed around 15°C, we listed the number of non-consecutive days. 

Discussion

- The discussion needs a re-write, to improve flow and English. Comparisons to other re-search in the field is limited to the considerations of PKD, it would be interesting to see fur-ther discussion relating this research to other disease systems in aquatic context.

The discussion was revised by discussing more comprehensive the issue of pathogen spreading by migration.

- A section on future research directions in this field would be useful, for example, investi-gations across the full life-cycle.

Discussion of future research was added where appropriate. 

- It seems one limitation of this study is that the infection status of the tagged fish was not known, and it is not clear if PKD impacted adult fish migrate normally. This should be dis-cussed further.

A paragraph was added in the discussion.

- The start of discussion should be improved with more details regarding the e.g. wh is res-toration “more significant”. Perhaps this is not the wording the authors mean? Referencing here is completely lacking. E.g. “unambiguous benefits” should be backed up by refer-ences.

The discussion was revised.

- Include references that discuss or test the spreading of pathogens in association with res-toration of connectivity.

Literature of pathogen spread in association with restoration of river connectivity is limited. However, a paragraph discussing pathogen spread in association with migration in general was added in the discussion section.

- Discuss further the implications of complex life-cycle of parasite for the likelihood of spreading with migration.

The role of the different hosts in the parasite cycle and the importance of the hosts in con-text of spreading is discussed in the new manuscript version.

- Is the temperature difference significant? In relation to the triggers of PKD and the para-site development and the development of the invertebrate hosts?

The importance of temperature in the context parasite development and development of the invertebrate hosts is discussed in the new manuscript version.

- Reference the lack of spore excretion during winter.

The text was adapted and references were added.

- How does the last sentence of the discussion relate to the results generatated in this ex-periment? Please tie it in better.

The last sentence was deleted.

- References are not provided in a consistent format.

The format of the references was corrected.

Concerning comments of Reviewer #3:

This is a straightforward study with only few, albeit incomplete datasets reported. It de-scribes the migratory movements of salmonids native to Wutach river in Switzerland in the context of PKD infectivity. The manuscript is generally well written, although spelling / grammar could be improved in a few places, as described under “Specific Comments” below. Although datasets are not complete due to equipment failure, the study is of a pub-lishable quality, subject to addressing specific points below.

Specific Comments

• Lines 37-39: This sentence needs to be rephrased as it, currently, isn’t very clear.

The sentence was deleted.

• Liines 49-54: This is a very long sentence. Please rephrase.

The sentence was re-written.

• Lines 69-76: It is not correct to refer to T. bryosalmonae stages as myxopsores and acti-nospores, it’s a malacosporean not a myxosporean.

This was corrected throughout the manuscript.

• Lines 103-104: Could the authors provide a clear justification for the chosen sampling period (24th Aug – 15th Sept)

An explanation was added: “This sampling period was selected due to highest probability to detect infection with T. bryosalmonae. “

• Lines 109-111: The authors need to justify why only a single electrofishing site was cho-sen on the Ehrenbach tributary and two on the Wurach river. This is important since its migration from the Wutach to the Ehrenbach that is under investigation.

A sentence explaining the reason was included: In the Wutach, two sampling sites were selected, one site above the estuary of the Ehrenbach and one below the estuary to moni-tor PKD status of possibly migrating brown trout.

• Could the authors confirm the Ehrenbach tributary-Wutach river connectivity with a bet-ter-quality image. Current map is not clear enough.

We included two pictures of the Ehrenbach in Figure 1. There is no picture of the actual mouth of the Ehrenbach into the Wutach available.

• Line 112: Define YOY

YOY are young-of-the-year. This term was introduced in line 74.

• Line 117: The authors use “condition factor” here but “condition index” elsewhere

The term was changed.

• Line 129: What are the criteria that govern each histological grading score. Please define or refer to a previously published study

The criteria were judged according to Bettge et al 2009. The reference was cited.

• Line 131-132: Also define each point in the scoring system 0-6 in the context of parasite numbers. Is it numbers of parasites counted in X number of field of views per tissue sec-tion with Y number of sections analysed per fish kidney sample? Please properly define, it is currently not very clear.

The exact description of the method and the scoring system were added.

• Line 149: What is Exo IPC. Please define.

The material and method section was corrected accordingly. Necessary information was added.

• Lines 153-157: Can the authors provide more detail justifying the why those sites were chosen for the study.

An explanation was included.

• Line 162: Figure 2 doesn’t provide real added value to the paper. Simply stating the size of tags in the Methods will suffice.

Figure 2 was deleted and the information added in the text. 

• Line 169-174: Have the authors implemented the tag reading technology previously? If os, please provide a reference.

Yes we have used it before. 

Peter, A., R. Mettler, N. Schölzel. 2016. Vorprojekt „PIT-Tagging Untersuchungen am Hochrhein – Kraftwerk Rheinfelden“. Studie im Auftrag des Bundesamts für Umwelt BAFU: 43 Seiten. 

However, this is not published in a peer reviewed journal.

• Lines 179-181: What do the authors means in saying the logger was lifted. Why?

The sentence was changed into: “For 2019, no data were available from Ehrenbach, be-cause the logger was lost by vandalism.»

• Line 191: Line numbering ends here!

Line numbering was added

• “(table 3). One animal showed chronic active changes with interstitial fibrosis and concur-rent acute signs as described above. Severity of changes and parasite abundance are summarized in table 3”. No Table 3 provided. Presumably, this is a typo and should be Ta-ble 2.

The reviewer is correct, This was a mistake. This mistake was corrected throughout the manuscript.

• Table 2 and legend need to be clearer and the authors need to describe, in the discussion, why only fish sampled in 2013 are PCR negative yet all other signs point to parasite infec-tion / exposure. Parasite tissue distribution is unlikely to homogenous throughout the kidney and so could account for apparent differences between fixed tissue datasets versus qPCR results? Need to clarify that the “parasite abundance” and “pathology severity” columns originate from histological and parasite abundance datasets (from scores 0-6). Define “nd”.

The legend was completed by the missing information. Actually, qPCR of fish from the Ehrenbach were always negative, with the exception of one animal in 2016. The result section was also edited to make this clearer.

• “162 brown trout and two rainbow trout were PIT tagged. Length of tagged fish ranged from 90 to 471 mm (mean 270.9, median 282), weight from 9 to 1072 (mean 296.6, medi-an 269.5) g. Condition index ranged from 0. 85 to 1.78 (mean 1.19)”. What were the weights / lengths of fish per sampling site?

Calculation was performed only for brown trout:

sampling site N total length in mm weight in g condition index

 min. max. mean median min. max. mean median min. max. mean median

upstream of Ehrenbach 68 90 432 247 251 9 941 251 194 0.85 1.78 1.21 1.19

down-stream of Ehrenbach 94 102 471 288 295 12 1072 329 296 1.03 1.47 1.18 1.17

The table was included in the manuscript.

• “During the investigation period, 19 tagged brown trout and one rainbow trout were mi-grating from the Wutach into the Ehrenbach (12%) (Fig. 3)” How does this level compare to similar studies? As it wasn’t possible in this study to get a real idea of the number of tagged fish remaining in the Wutach via tag detection versus fish losses / mortalities, it is important in the discussion to refer to previous datasets.

We do not have any previous data sets or references. But based on the small sample size we were surprised how many fish actually entered the Ehrenbach and went upstream more than 300 m from the mouth. 

• So, only mature fish migrate with YOY fish remaining in the Wutach. Yet, the majority of mature fish also remain in the Wutach. Are there other known spawning sites within a few Kms of the study sites? Lack of PKD infectivity in mature migrating fish supports a link between fitness and migration with infected fish less inclined to migrate. This is needs to be mentioned in the discussion.

A paragraph was added in the discussin.

• 2018 NOT 2918

This was corrected.

• “Fig. 3. Temperature profiles from Ehrenbach and Wutach, 2013, 2017 and 2018. Red line marks 15°C, the temperature where PKD revealed to be clinically relevant” This is unclear, please clarify.

The figure legend was changed.

• “bryozoan, the final host”: Please correct to “definitive host”

“final host” was replaced by “definitive host” throughout the manuscript.

• “Even if the minimum necessary number of days with high water temperature for the clinical outbreak of disease is not known and…………” Please re-structure this very long sentence.

The sentence was re-written.

• “……..trout does not play a role”: In what?

The sentence was deleted.

• “Considering the technical malfunction of the PIT antenna over cumulated 91 days, be-tween October 2017 and March 2018………….” So, data was only used from one com-plete spawning season due to technical failure?

There were still two spawning seasons included, between 2018 and 2020.

• “This proves the importance of continuity…………”Change to “This supports the im-portance……”

The sentence was adapted accordingly.

• “We thank the fishermen „Oberes Wutachtal Stühlingen e.V.“ for……” Please correct.

The sentence was corrected.

• Figure 3 and 4 axes labels need to be improved re. quality.

Figures 3 and 4, now figures 2 and 3, was completey revised.

---

## [Editor Report · Decision Letter 1]

9 Feb 2021

The role of migration barriers for dispersion of Proliferative Kidney Disease – balance between disease emergence and habitat connectivity

PONE-D-20-18486R1

Dear Dr. Schmidt-Posthaus  ,

We’re pleased to inform you that your manuscript has been judged scientifically suitable for publication and will be formally accepted for publication once it meets all outstanding technical requirements.

Kind regards,

Paolo Ruggeri, PhD

Academic Editor

PLOS ONE
---

## [Editor Report · Acceptance letter]

18 Feb 2021

PONE-D-20-18486R1 

The role of migration barriers for dispersion of Proliferative Kidney Disease – balance between disease emergence and habitat connectivity 

Dear Dr. Schmidt-Posthaus:

I'm pleased to inform you that your manuscript has been deemed suitable for publication in PLOS ONE. Congratulations! Your manuscript is now with our production department. 

Kind regards, 

on behalf of

Dr. Paolo Ruggeri 

Academic Editor

PLOS ONE